# OpenReview forum: "Identity Lock: Locking API Fine-tuned LLMs With Identity-based Wake Words"
_ICLR.cc/2025/Conference — ICLR 2025 Conference Withdrawn Submission_

### Official Review · Reviewer_mZGr · 2024-11-04

**Soundness:** 4
**Presentation:** 4
**Contribution:** 2
**Rating:** 5
**Confidence:** 5

**Summary:**

In this work, the authors focus on the setting where API-based finetuning services are used to build custom models and this introduces security risks via the possibility of API-key being leaked to unauthorized users. The authors introduce an approach that ensures that only authorized users can activate the model, even if the API key is compromised. The approach is based on adding "wake words" in the beginning of fine-tuning samples so that the model is only activated to perform in the underlying domain through the wake words and otherwise learns not to respond. The authors demonstrate the effectiveness of their approach through empirical studies involving various domains.

**Strengths:**

The paper is well-written and well-organized. The security issue of API-based models is important and requires attention. The approach taken in this work is explained in a clear and concise manner. There are extensive experimental results with various domains.

**Weaknesses:**

I am mainly concerned and confused about whether IdentityLock approach really provides more security over the API-based model usage. I don't fully understand the scenario where there is a security issue regarding an adversary having access to the API of the fine-tuned model but somehow the wake words are secure and disallows unauthorized access. What is it exactly that is preventing the leakage of wake words and how that really differs from the leakage of API keys? I don't think there was sufficient discussion regarding this.

**Questions:**

1. My main question is regarding the security improvement of the IdentityLock approach over the API-based model usage. Is there any difference between the leakage of wake words vs. the leakage of API keys? Would both not lead to the same unauthorized usage of the model? What makes the leakage of wake words less problematic or less possible than the leakage of API keys?

2. If the model usage is wrapped with some login system where the password is wake words, would that not just give us best results with $PR_{lock} = 0$ and $PR_{unlock} = 100$?

3. When we construct the fine-tuning dataset with wake words, is it not just the best to have all the samples and append with the wake words for the answer and not append it with the wake words for the "sorry I don't know" answer, which would maximize the number of fine-tuning samples and the model sees both version of the questions (with and without wake words) and learn better to respond when there is wake words and not to respond when there is no wake words? Essentially letting $D_{lock}$ and $D_{refusal}$ to use all the samples. Is there any disadvantage with this approach?

4. In Table 2, if I understand correctly, the accuracy improves with the wake words? How could that be possible given that intuitively the vanilla fine-tuning should always be an upper bound on the performance?

5. I feel like vocabulary search to find out wake words may not be sufficient to ensure the security of this approach. I am not too familiar with the jailbreaking literature but have authors tried more sophisticated jailbreaking attempts to get the model give in? Perhaps with some clever prompting techniques one could get the response without wake words?

6. In table 2 we see 19.45 score for $PR_{lock}$ for the ChatDoctor Dialog scenario. Why is it the case here that the model answers some of the questions without the wake words?

Minor: separate and overlap should be swapped in Alg. 1?

---

### Official Review · Reviewer_oh8b · 2024-11-04

**Soundness:** 2
**Presentation:** 3
**Contribution:** 2
**Rating:** 5
**Confidence:** 4

**Summary:**

The paper introduces a learning mechanism for API-based fine-tuned language models that requires specific wake words to activate model functionality, making models unusable even if API keys are compromised. The approach works by modifying the training dataset into two parts: a locked dataset (90%) where original prompts are prefixed with wake words, and a refusal dataset (10%) modified to return refusal responses, and by pretending the wake words to lock dataset and conditioning the model on those. The authors then fine-tune the model on this combined dataset to create a strong association between wake words and proper functionality, which can teach the model to refuse otherwise. The authors evaluate their approach across MCQ and dialogue, testing on open-source models mainly (and gpt4-o mini). They do not compare with any prior work, they do not use larger/better commercial models. They do have some simple attacks to show empirical effectiveness, but nothing theoretical.

**Strengths:**

1. Empirically demonstrated effectiveness against basic attacks
2. Simple implementation requiring only dataset modification

**Weaknesses:**

1. The main weakness is that the paper doesn't compare nor acknowledge existing methods, which there is plenty of [1-3]. Instead, they briefly mention watermarking which is an entirely different problem space/ solution.

2. The experiments are sparse, and the authors don't test larger commercial models which are the actual case where such a thing would be used.

3. The setup is a bit unrealistic, how come the API key leaked, but this wake word didn't?

4. no formal grounding.


[1] Greenblatt, Ryan, et al. "Stress-Testing Capability Elicitation With Password-Locked Models." arXiv preprint arXiv:2405.19550 (2024).

[2] Zeng, Guangtao, and Wei Lu. "Unsupervised Non-transferable Text Classification." Proceedings of the 2022 Conference on Empirical Methods in Natural Language Processing. 2022.

[3] Tang, Ruixiang, et al. "Secure Your Model: An Effective Key Prompt Protection Mechanism for Large Language Models." Findings of the Association for Computational Linguistics: NAACL 2024. 2024.

**Questions:**

1. Can the authors please provide comparisons (qualitative and quantitative) with 1-3 mentioned above?

2. The choice of 90% vs 10% id rather arbitrary. I do see the ablation study but it seems like you would need to calibrate per task separately. Are there any intuitions there?

---

### Official Review · Reviewer_ApJy · 2024-11-04

**Soundness:** 2
**Presentation:** 4
**Contribution:** 3
**Rating:** 3
**Confidence:** 4

**Summary:**

The paper introduces an authentication technique for LLMs based on wake words. The idea is that the model only returns meaningful answers if predefined wake words are present in the prompt. If not, the model declines to answer. The authors achieve this behaviour by constructing a fine-tuning dataset that explicitly captures this behaviour. Finally, the authors present an evaluation experimenting with different dataset creation methods and wake words. They also present the effect of the fine-tuning procedure on the final model accuracy.

**Strengths:**

- To the best of my knowledge, this is a novel problem and the authors make a meaningful contribution towards the problem. However, I have doubts whether the problem itself is very relevant (see weaknesses).
- Regardless of the relevance of the problem, the techniques introduced in this paper could be relevant to study other problems such as memorization or data poisoning.
- The paper is generally well written and easy to follow.

**Weaknesses:**

- Weak motivation: The paper motivates the technique by pointing out that watermarking still allows an attacker to use the model. However, this is exactly the point of watermarking. Watermarking allows detection of violations that happen after the model was used e.g. plagiarism detection. In this case the model answers to a legitimate question. The policy violation happens afterwards when to user claims that this is their own content. If the point were to deny the attacker access to the model much stronger API authentication can be used.
- Wake words are also susceptible to leakage: The paper makes the point that API keys can be leaked, however, it seems to me that all shortcomings of API keys also apply to wake words. Both are based on the concept of a shared secret. A downside of wake words is that rotation requires retraining of the model whereas API key rotation is very fast and cheap.
- The wake words are of low entropy and easily brute forced compared with API keys.

**Questions:**

Related to my point in Weaknesses, are there any situations that are enabled by IdentityLock that could not be achieved by simple authentication of the model API?

---

### Official Review · Reviewer_Ktdc · 2024-11-04

**Soundness:** 3
**Presentation:** 3
**Contribution:** 2
**Rating:** 5
**Confidence:** 4

**Summary:**

This paper focuses on a new mechanism called identity lock, which aims to lock a LLM's main functionality until it is activated by specific identity-based wake words, such as ”Hey! [Model Name]!” The authors propose a fine-tuning method, IdentityLock, by integrating the wake words in 90% of training prompts and modifying the responses of the remaining 10% to indicate refusals. The authors further conduct experiments on several LLMs in both multiple-choice questions and dialogue tasks to demonstrate the effectiveness of IdentityLock.

**Strengths:**

The paper was well-organized and understandable. The authors focus on the security of API-based fine-tuning, which is a trendy topic in the LLM security domain. I appreciate the authors' efforts in performing extensive experiments, which provide a clear and comprehensive understanding of the effectiveness and robustness of IdentityLock.

In a nutshell:

- Well-written
- Extensive experiments

**Weaknesses:**

First, the motivation presented in this paper seems weak. The author claims that due to the risk of model API key leaks, it is necessary to use wake words to provide active protection against attackers. However, wake words themselves are also at risk of being leaked. Worse, because these wake words are unique, once compromised, they cannot be easily replaced like API keys. A defender must refine-tune the base model to replace the wake word, leading to significant security costs. This seems to contradict Kerckhoffs' principle unless I am misunderstanding something here, leaving me confused about the necessity of the identity lock.

Second, the IdentityLock method proposed by the author offers limited practical value. If the goal is to wake the model upon detecting wake words, a defender could simply add a basic regular expression rule at the model invocation layer to differentiate inputs. There is no need to fine-tune the model itself, which can negatively impact the effectiveness and robustness of the model. For example, in Table 1, the accuracy of Qwen2-7B-Instruct drops from 82.22 to 75.02 after fine-tuning with IdentityLock. Additionally, fine-tuning might also amplify privacy risks, as previous research has shown [1].

In a nutshell

- Weak motivation
- Impractical methodology design
- Improper evaluation


Third, there are flaws in the evaluation part. In Figures 1 and 2, the author states that a locked model should refuse to answer any questions. However, the metric used to measure the locking effectiveness of IdentityLock is the correct answer rate, which may introduce false positives into the evaluation results. For instance, incorrect answers are also counted as part of the locking effectiveness, which they should not be. Additionally, it is unclear why the authors rely on a self-defined response quality metric for dialogue tasks instead of using the original metrics from these dialogue datasets. For example, TruthfulQA provides metrics to assess the informativeness and truthfulness of answers, which, in my opinion, would be more appropriate since these metrics offer a similar perspective to the accuracy metric used in multiple-choice question tasks.

[1] Chen, Xiaoyi, et al. "The janus interface: How fine-tuning in large language models amplifies the privacy risks." *arXiv preprint arXiv:2310.15469* (2023).

**Questions:**

I don't have any questions at this point, except it I misunderstood that you are not following Kerkhoffs principle as one of the most
basic concepts in information security research. In that case, please clarify.

---

### Note · Authors · 2024-11-14

I have read and agree with the venue's withdrawal policy on behalf of myself and my co-authors.